# Quantitative structural mechanobiology of platelet-driven blood clot contraction

Oleg V. Kim[1,2,3,4], Rustem I. Litvinov[1,5], Mark S. Alber[2,4,6] & John W. Weisel[1]

Blood clot contraction plays an important role in prevention of bleeding and in thrombotic disorders. Here, we unveil and quantify the structural mechanisms of clot contraction at the level of single platelets. A key elementary step of contraction is sequential extension–retraction of platelet filopodia attached to fibrin fibers. In contrast to other cell–matrix systems in which cells migrate along fibers, the "hand-over-hand" longitudinal pulling causes shortening and bending of platelet-attached fibers, resulting in formation of fiber kinks. When attached to multiple fibers, platelets densify the fibrin network by pulling on fibers transversely to their longitudinal axes. Single platelets and aggregates use actomyosin contractile machinery and integrin-mediated adhesion to remodel the extracellular matrix, inducing compaction of fibrin into bundled agglomerates tightly associated with activated platelets. The revealed platelet-driven mechanisms of blood clot contraction demonstrate an important new biological application of cell motility principles.

[1] Department of Cell and Developmental Biology, University of Pennsylvania School of Medicine, Philadelphia, PA 19104, USA. [2] Department of Applied and Computational Mathematics and Statistics, University of Notre Dame, Notre Dame, IN 46556, USA. [3] Harper Cancer Research Institute, Notre Dame, IN 46556, USA. [4] Department of Mathematics, University of California Riverside, Riverside, CA 92505, USA. [5] Institute of Fundamental Medicine and Biology, Kazan Federal University, Kazan 420008, Russia. [6] Department of Medicine, Indiana University School of Medicine, Indianapolis, IN 46202, USA. Correspondence and requests for materials should be addressed to M.S.A. (email: malber@ucr.edu) or to J.W.W. (email: weisel@mail.med.upenn.edu)

Contraction of blood clots and thrombi is an inter-disciplinary problem related to fundamental aspects of cell biology, including cell motility and interaction of cells with extracellular matrix, as well as to blood clotting and its disorders, such as heart attack, stroke, and venous thromboembolism. Platelet-driven clot contraction is important for hemostasis and wound healing as well as for restoring the blood flow past otherwise obstructive thrombi within a vessel[1]. In a more general context, the ability of cells to contract is an essential biological function of various biological systems, including muscle cells, endothelial cells, hepatic stellate cells, fibroblasts, and activated platelets utilizing the same intracellular contractile protein machinery[2–5]. Non-muscle myosin IIA is critical for platelet contraction by interacting with actin to form a contractile unit similar to other actomyosins in cell motility. The platelet integrin $\alpha_{IIb}\beta_3$ forms a transmembrane link between fibrin outside the platelet and actin inside the platelet[6, 7] connected to the integrin via talin[8–10].

The studies described so far define the components necessary for clot contraction, but the physical mechanism has still been unknown. While it has been demonstrated that platelets and fibrin are necessary for contraction of clots, which has been studied at different special scales from a whole clot to the single-cell level[11–16], much less is known about how individual platelets or small platelet aggregates exert contractile force on individual fibrin fibers and how this tension causes collapse of the entire filamentous network and reduction of clot volume. To get insights into the structural reorganization of the extracellular matrix underlying platelet-driven clot contraction biomechanics, we use high-resolution confocal microscopy and rheometry to perform concurrent three-dimensional (3D) dynamic structural and mechanical measurements of the platelet-fibrin meshwork over the course of clot contraction. We pay special attention to the elementary steps of clot contraction in the real-time scale by visualizing single contracting platelets bound to an individual fibrin fiber and their effects on remodeling of the entire fibrin network powered by multiple contracting platelets. We discover a structural mechanism by which local platelet-fibrin interactions result in dramatic modifications of the whole clot architecture.

## Results

**Platelets bend and shorten individual fibrin fibers.** To determine the biomechanical mechanisms that drive contraction of blood clots, we performed time-lapse (50 min) high-resolution z-stack imaging of the contracting platelet-fibrin meshwork, which enabled us to watch spatio-temporal structural rearrangements of single fibers attached to individual platelets or their small aggregates ($\sim 10\,\mu m$) during the course of clot contraction.

When an adherent platelet spread over a single fibrin fiber, its filopodia stretched along the fiber axis (Fig. 1a). Initially, the fiber remained stationary until the filopodium began to contract, at which point it was pulled by the filopodium, such that the attached fiber was bent sharply, so that it formed a kink and moved toward the main part of the cell. Finally, the displaced fiber became deformed, crumpled and after about $t = 700\,s$ got incorporated into a bundle of compacted fibrin fibers (Fig. 1a). Subsequently, another filopodium bound to the same fiber and pulled on it, such that the platelet pulled with a hand-over-hand motion, like pulling on a rope. To quantify the displacement of fibrin fibers powered by platelet filopodia, we measured the length of each kink of individual fibrin fibers and the rate of fiber shortening (Fig. 1b, c). The average length of fibrin kinks ($1.6 \pm 0.5\,\mu m$, mean $\pm$ SD, $n = 42$) was $\sim 1.7$ times smaller than the average length of filopodia (Fig. 1c) ($2.7 \pm 1.1\,\mu m$, $n = 300$), indicating that each filopodium contracted partially (by a part of

its length). As a result of filopodia pulling, platelet-attached fibrin fibers shortened (Fig. 1b, inset) at a mean rate of $0.003 \pm 0.002$ $\mu m/s$ ($n = 30$). The dynamic interaction of a platelet and a single fibrin fiber is shown in the Supplementary Movies 1, 2.

The observed platelet-fibrin dynamic rearrangements have substantial differences from the behavior of fibroblasts moving on and reshaping collagen in the extracellular matrix[17]. Although fibroblasts also pull hand-over-hand on collagen fibers and kink them, platelets use thin and short filopodia, while fibroblasts bind to collagen via larger protrusions or lamellipodia and do not generally contract the matrix[18]. In contrast to fibroblasts, platelets do not release the displaced fibers, rather remaining attached to them and compacting fibrin fibers locally into bundles and clusters. In addition, the force field generated by platelets was shown to be isotropic[19], whereas fibroblasts generate an anisotropic force field, which enables them to migrate actively[17]. Other additional differences can be found in Supplementary Note 1. Therefore, the platelet-fibrin interactions comprise distinct and understudied cell motility and matrix remodeling mechanisms.

**Platelets cause compaction of the fibrin network.** In addition to single cell/single fiber interactions, we observed larger scale fibrin-platelet associations (over $\sim 100\,\mu m$), in which a platelet aggregate attached via filopodia to multiple fibers (Fig. 1e, Supplementary Movie 3). Remarkably, platelets were able to stretch filopodia across the gap between the main body of the cell and the neighboring fibers and pull on the fibers transversely, re-orienting them toward the cell. Contraction of the filopodia caused the fibers' displacement followed by their compaction and even bundling. Sequential extension-retraction of platelet filopodia resulted in accumulation of the fibrin mass that was progressively more and more co-localized with the platelets. Similar to contraction of a single fiber, the platelet(s) remained attached to the bundle of compacted fibrin fibers with multiple fibers radiating outward from the cell or cell aggregate.

Filopodia extension and retraction can be considered as the elementary event of continuous clot contraction. Repetitive extension and retraction of filopodia caused an increasing fiber displacement and progressive accumulation/compaction of fibrin fibers co-localized with platelets. Thus, the main biomechanical mechanism of fibrin network remodeling is caused by the platelet filopodia-driven extension/retraction steps. To quantify the dynamics of fibrins compaction, we assessed the area of fibrin co-localized with platelets at different time points (Fig. 2). The mean total area of platelet-attached fibrin increased dramatically during first 30 min of contraction, indicating a 10-fold increase compared to the initial amount of platelet-associated fibrin (Fig. 2b). The average area of platelet-bound fibrin after normalization per platelet also increased with time and reached on average $5.4\,\mu m^2/cell$ after 40 min of contraction. Inhibition of platelet contractile proteins by the myosin inhibitor blebbistatin significantly reduced fibrin compaction and resulted in only a twofold increase of the mean total area of platelet-co-localized fibrin, reaching $2.5\,\mu m^2/cell$ (Fig. 2c). Blockage of $\alpha_{IIb}\beta_3$-fibrin interaction with abciximab precluded fibrin compaction (Supplementary Fig. 1).

Both single platelets and platelet aggregates actively performed fibrin remodeling. To see if platelet aggregation affected fibrin remodeling, we measured the fraction of fibrin co-localized with single platelets and with platelet aggregates (Fig. 2d). During clot contraction, the fraction of fibrin bundles compacted by platelet aggregates gradually increased from zero and reached $33 \pm 8\%$ ($n = 60$) at the final stage of contraction, while the relative fibrin mass that occupied an area smaller than a single platelet

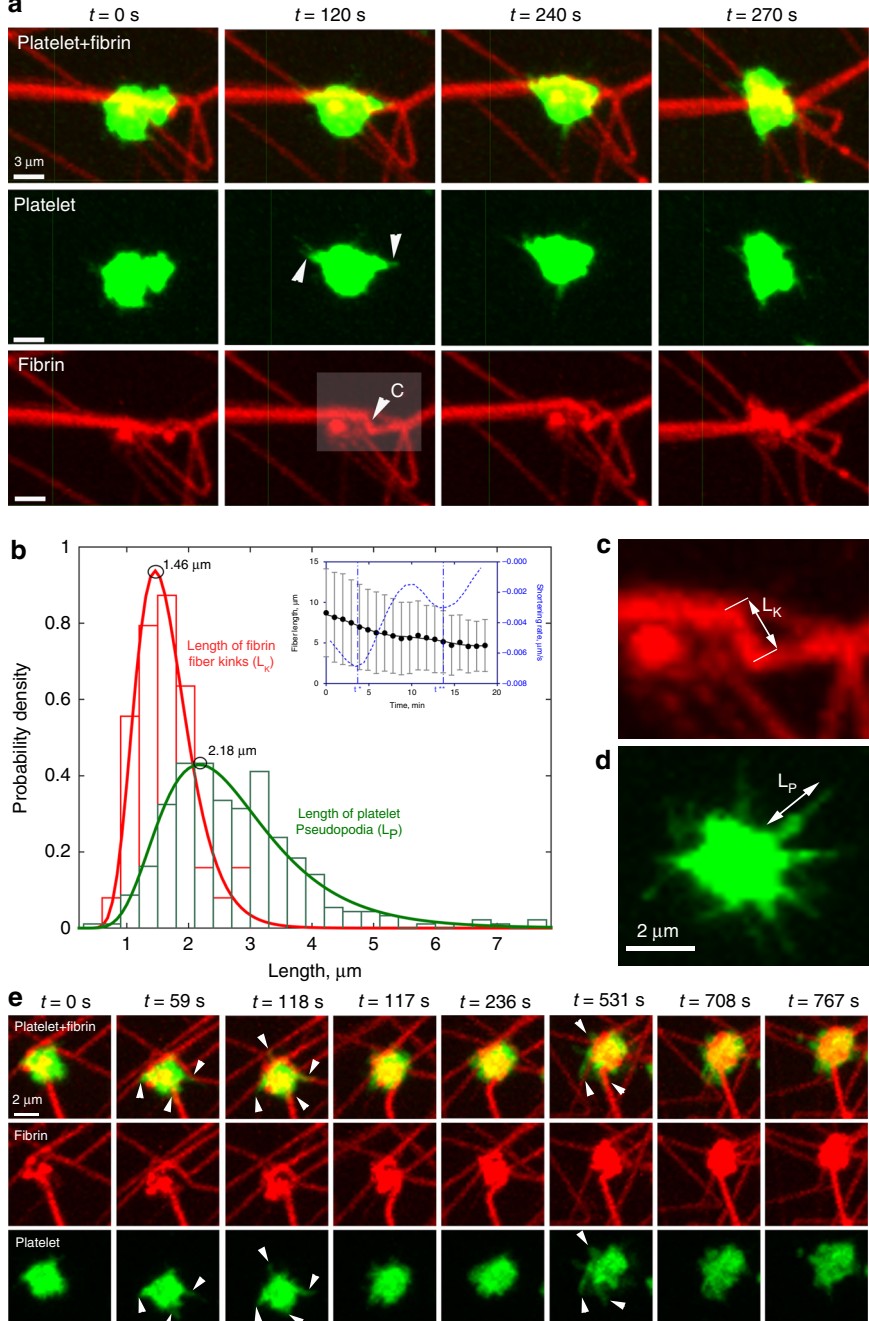

**Fig. 1** Time-lapse images of contracting platelets that cause bending, kinking, and local accumulation of a single fibrin fiber. **a** *Top row*: a platelet or a small platelet aggregate (*green*) attaches to a fiber (*red*) and spreads filopodia along the fiber axis that contract, inducing a fiber kink and pulling the fiber, compacting it into a dense fibrin knot or coil. For details see Supplementary Movie 1. **a** *Middle row*: platelet transformations, including attachment of filopodia to a fiber, spreading and contraction (corresponding to **a**, top row). **a** *Bottom row*: platelet-induced structural changes in a fibrin fiber. The *inset* ($t = 120$ s) shows formation of a kink. Arrows indicate filopodia attached to a fiber, including the kink. **b** Length distributions of the fiber kinks, $L_K$ ($n = 42$) and platelet filopodia, $L_P$ ($n = 300$). The *bars* represent experimental numbers and the curves are log-normal fits. *Inset*: shortening of a fibrin fiber (*black dots*) and the shortening rate (*blue dotted line*) caused by platelet contraction (mean ± SD, $n = 30$). $t^*$, $t^{**}$ are the microscopic phase transition times separating different regimes of filopodia shortening. **c** A zoomed fiber kink of a length $L_K$. **d** Filopodia with lengths defined as $L_P$; both parameters presented in **b**. **e** Serial images of a contracting platelet reveal reorganization and compaction of fibrin fibers surrounding the cell. *Top row*: combined fibrin and platelet images; *middle row*: fibrin network only; *bottom row*: platelet only. Arrows indicate the platelet's filopodia attaching to and pulling on fibrin fibers. (see Supplementary Movie 3 for the full sequence at high magnification and Supplementary Movie 4 for an individual platelet filopodium undergoing contraction)

($S^{SP} = 9 \, \mu m^2$) dropped from 100% to $33 \pm 9\%$ ($n = 110$). As the maximum size of a compacted fibrin bundle is determined by the size of an associated platelet aggregate, our findings suggest that platelet aggregation enhances remodeling of fibrin matrix and results in formation of larger clumps of densified (or compacted)

fibrin as compared to fibrin accumulated by single platelets. Since formation of compacted fibrin is driven by platelet actomyosin contractile machinery, a platelet aggregate dissipates more energy on pulling and compacting fibrin than an individual platelet by deforming a larger number of fibrin fibers. Therefore, platelet

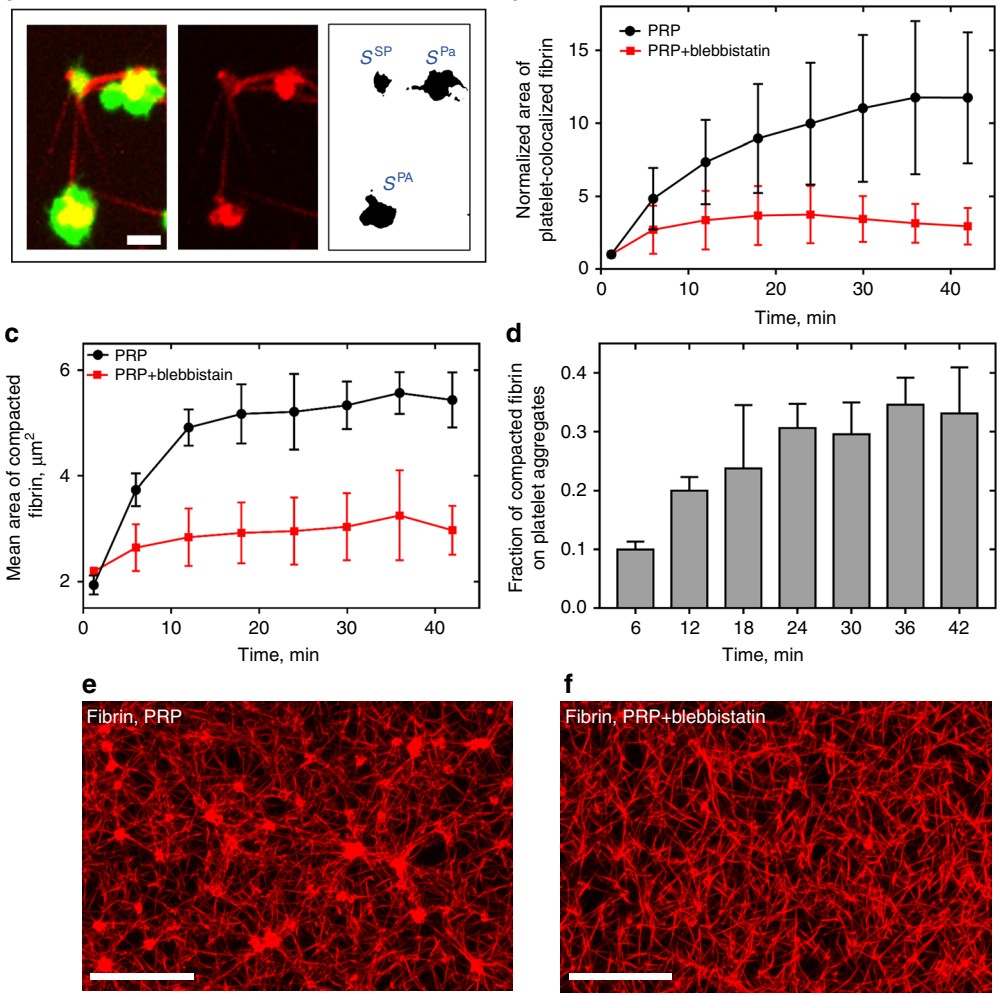

**Fig. 2** Platelet-induced fibrin compaction. (**a**, *left*)**:** a confocal image of platelets (*green*) and fibrin (*red*); (**a**, *center*): fibrin fibers and patches (compacted fibrin); (**a**, *right*): fibrin co-localized with platelets. Compaction of fibrin by a single cell ($S^{SP}$) and by two platelet aggregates ($S^{PA}$) is shown; *scale bar*: 3 μm. **b**, **c** Changes in the total area of platelet-associated fibrin normalized to its initial value **b** and the mean absolute area of platelet-co-localized fibrin (**c**) during the course of clot contraction (mean ± SEM). **d** Relative portion of fibrin material compacted by platelet aggregates, mean ± SEM, $n > 200$. The analyzed volume size is 217 μm × 217 μm × 34 μm. **e**, **f** Confocal images of a fibrin network in a PRP-clot formed and allowed to contract in the absence (**e**) and presence (**f**) of blebbistatin (300 μM). *Scale bar*: 30 μm

aggregates become more efficient in fibrin remodeling at the later stages of clot contraction forming large patches of compacted fibrin.

**Secondary clustering and fusion of fibrin-attached platelets.** Examination of the time-lapse confocal images of the contracting platelet-fibrin meshwork revealed secondary platelet clustering as a result of close spatial approximation of the neighboring fibers that bear attached platelets (Fig. 3a, Supplementary Movies 4, 5. As soon as platelets started to compact or densify fibers by pulling on them, either longitudinally or transversely, the distance between the fibrin-attached cells shortened and platelets eventually merged into a larger cellular clump or aggregate (Supplementary Fig. 2).

To quantify the phenomenon of platelet clustering, we examined the distribution of platelet-specific fluorescent areas in the z-projection obtained from 3D confocal time-series of the contracting platelet-fibrin meshwork. Although the peak of the platelet size distribution (~6 μm²) did not change during contraction (Fig. 3b), the larger cell associations in the tail of the histogram (Fig. 3b, inset) displayed an increase of average size

(Fig. 3c) from 30.2 ± 3.6 to 40.9 ± 18.8 μm² ($P = 0.008$, a two-tailed Mann–Whitney test). This secondary enlargement of platelet aggregates incorporated into fibrin likely results in generation of a higher contractile force that would promote clot contraction at the later stages.

**Structure-based kinetics of clot contraction.** Next, we correlated the time-dependent structural changes with the overall kinetics of clot contraction. Vertical z-projections of confocal images of a fibrin network (186 μm × 186 μm × 30 μm) collected during clot contraction indicated that fibrin density increased with time (Fig. 4a, Supplementary Movies 6–8), as measured by the fibrin fluorescence intensity per square unit (Fig. 4b). This kinetic curve analyzed using the first derivative, or velocity, revealed three distinct phases in clot contraction named "pre-contraction" or lag period (phase $P_1$), "active contraction" (phase $P_2$), and "final contraction" (phase $P_3$), each characterized by duration and a rate constant. The average duration of $P_1$ was $t_1 = 5.0 ± 1.2$ min, $P_2$ lasted $t_2 = 28.0 ± 1.2$ min followed by $P_3$ with $t_3 > 30$ min. The corresponding rate constants were $k_1 = 1.07 ± 0.05$/s; $k_2 = 0.01 ± 0.005$/s; $k_3 = 0.004 ± 0.002$/s.

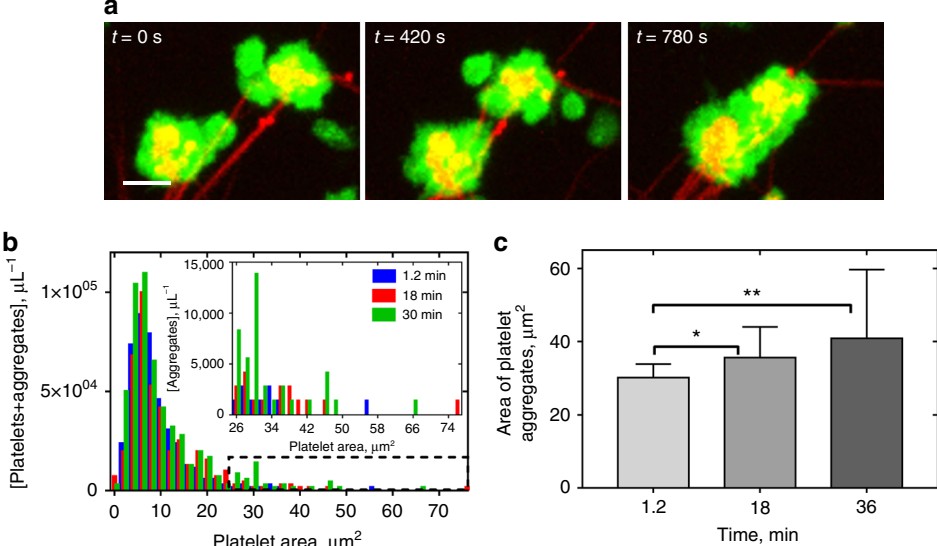

**Fig. 3** Clustering of the fibrin-attached neighboring platelets. **a** Serial confocal images showing formation of secondary platelet clusters due to approximation of platelet-bearing fibrin fibers during clot contraction. Platelets are *green* and fibrin is *red*. Scale bar: 5 μm. For the entire sequence, see Supplementary Movies 5, 6. **b** Platelet area distribution in a contracting clot at different time points with the inset showing the distribution of the larger platelet clusters. **c** Changes of the average platelet cluster size at different time points of clot contraction corresponding to contraction phases $P_1$, $P_2$, and $P_3$ (mean ± SD, $n > 22$, *$P < 0.05$, **$P < 0.01$, a two-tailed Mann−Whitney test)

To determine the effects of inhibition of myosin IIA and fibrin-$\alpha_{IIb}\beta_3$ binding on the observed kinetics, we examined changes in clot contraction in the presence of blebbistatin, a selective inhibitor of myosin II, or abciximab, a specific antagonist of the platelet integrin $\alpha_{IIb}\beta_3$. Blebbistatin caused a decrease in the rate and extent of fibrin densification, which dropped several-fold at the final phase of contraction (Fig. 4c, d). The delayed network densification of the fibrin network correlated with significantly reduced compaction of fibrin fibers (Fig. 2d-f). Addition of abciximab to the contracting clot also dramatically reduced the degree of fibrin densification and no compaction of fibers by platelets was observed (Supplementary Fig. 1).

**Mechanics-based kinetics of clot contraction**. The integral structural changes of the contracting fibrin network were correlated with dynamic measurements of the contractile stress generated by the entire clot. The stress generated by the clot was measured in a rheometer as the stress on the two horizontal plates during isometric contraction (Methods section). The time course of changes in the contractile stress largely concurred with the time-dependent structural changes. A very gradual initial increase in the contractile stress turned to asymptotic behavior at the end of contraction (Fig. 4e, Supplementary Movies 9−11). However, the kinetic analysis using time peaks of the first derivative revealed four distinct phases with the second ($P_2^A$) and third ($P_2^B$) sub-phases showing up as one phase ($P_2$) in the densification curve (Fig. 4f). The borders between the phases were at $t_1 = 3.9 \pm 0.1$ min, $t_2 = 11.8 \pm 0.1$ min, $t_3 = 31.5 \pm 0.1$ min, which were close to the transition times observed for fibrin densification (Fig. 4b). The corresponding compression or loading rates were $5.6 \pm 1.3$ Pa/min, $4.3 \pm 0.3$ Pa/min, $3.0 \pm 1.2$ Pa/min, and $0.6 \pm 0.3$ Pa/min. Remarkably, the single-fiber contraction rates (Fig. 1b, inset) signify local minima around $t^* = 4$ min and $t^{**} = 14$ min, which correspond to the $P_1$−$P_2$ and $P_2^A$−$P_2^B$ phase transition times obtained from macroscopic whole clot kinetics measurements[20]. Because platelet contraction and fibrin network remodeling are the key elements of the entire process of clot contraction, the observed correspondence between the phases

determined at the micro-scales and macro-scales suggests that the multiphasic behavior observed at the whole clot level originates from the time-dependent functional mechanisms at the level of single platelet/single fiber interactions.

The mechanical properties of the contracting clots were also affected by the presence of myosin II and $\alpha_{IIb}\beta_3$ inhibitors. Thus, inhibition of platelet myosin II attenuated the normal stress threefold, while blocking $\alpha_{IIb}\beta_3$-fibrin binding drastically reduced the contractile stress, which fluctuated around zero (Fig. 4e). The storage and loss moduli in the presence of abciximab were decreased significantly compared to non-inhibited clot contraction or inhibition of platelet myosin II (Fig. 4g, h).

The contractile force generated by a single platelet can be estimated as the integral force pulling on the rheometer plates divided by the number of platelets in the clot volume. Such a calculation yielded a force of ~ 0.3−1.7 nN per platelet, which is in agreement with the forces 0.3−2.1 nN measured by others[12–14, 21]. These numbers are smaller than the force generated by a single platelet of 29−34 nN[15, 19]. This discrepancy may be attributed to the multi-directional force generated by platelets in bulk compared to a single cell, reorganization of actin in merging cells[22] or variations in cell response to matrix stiffness[23].

**Clot shrinkage and spatial non-uniformity of contraction**. To establish a connection between microscale cell motion and alterations of the fibrin network bulk structural properties, we examined the spatial displacement of single platelets and platelet ensembles over time during clot contraction. Tracking individual cells in confocal 3D stacks enabled us to reconstitute a 3D spatial-temporal displacement field and derive clot volume changes for each phase of contraction (Fig. 5). We assessed the deformation of the fibrin network by platelets in terms of the volumetric strain $\epsilon = \int_0^t \int_V \nabla \cdot \mathbf{u}\, dv\, d\tau$, where $\mathbf{u}$ is a platelet displacement rate field measured by tracking individual platelets incorporated into a 3D fibrin network, $t$ is the time of contraction, and $V$ is the volume of the clot domain.

The mean displacement rate ("platelet speed") decreased gradually with time and varied from $0.014 \pm 0.009$ μm/s at the beginning of contraction to $0.007 \pm 0.004$ μm/s at 40 min of clot

contraction (Fig. 5c). Inhibition of platelet myosin IIA-actin complex and blocking $\alpha_{IIb}\beta_3$ both resulted in reduction of the platelet moving speed down to its constant values $0.004 \pm 0.003$ µm/s and $0.005 \pm 0.002$ µm/s, respectively. The calculated volumetric strain for "pre-contraction" (phase $P_1$), "active contraction" (phases $P_2$), and "final contraction" (phase $P_3$) indicated a significantly higher volume decrease (up to 10 times) in the absence of inhibitors that reached $27 \pm 2.7\%$ during the final contraction phase ($P_3$) (Fig. 5d). In the presence of blebbistatin and abciximab, the clot volume decreased by $7.7 \pm 2.4\%$ and $4.2 \pm 0.3\%$, respectively (Fig. 5d). Thus, inhibition of $\alpha_{IIb}\beta_3$-mediated platelet-fibrin interactions by abciximab had a greater inhibitory effect on clot contraction dynamics than the suppression of myosin IIA by blebbistatin.

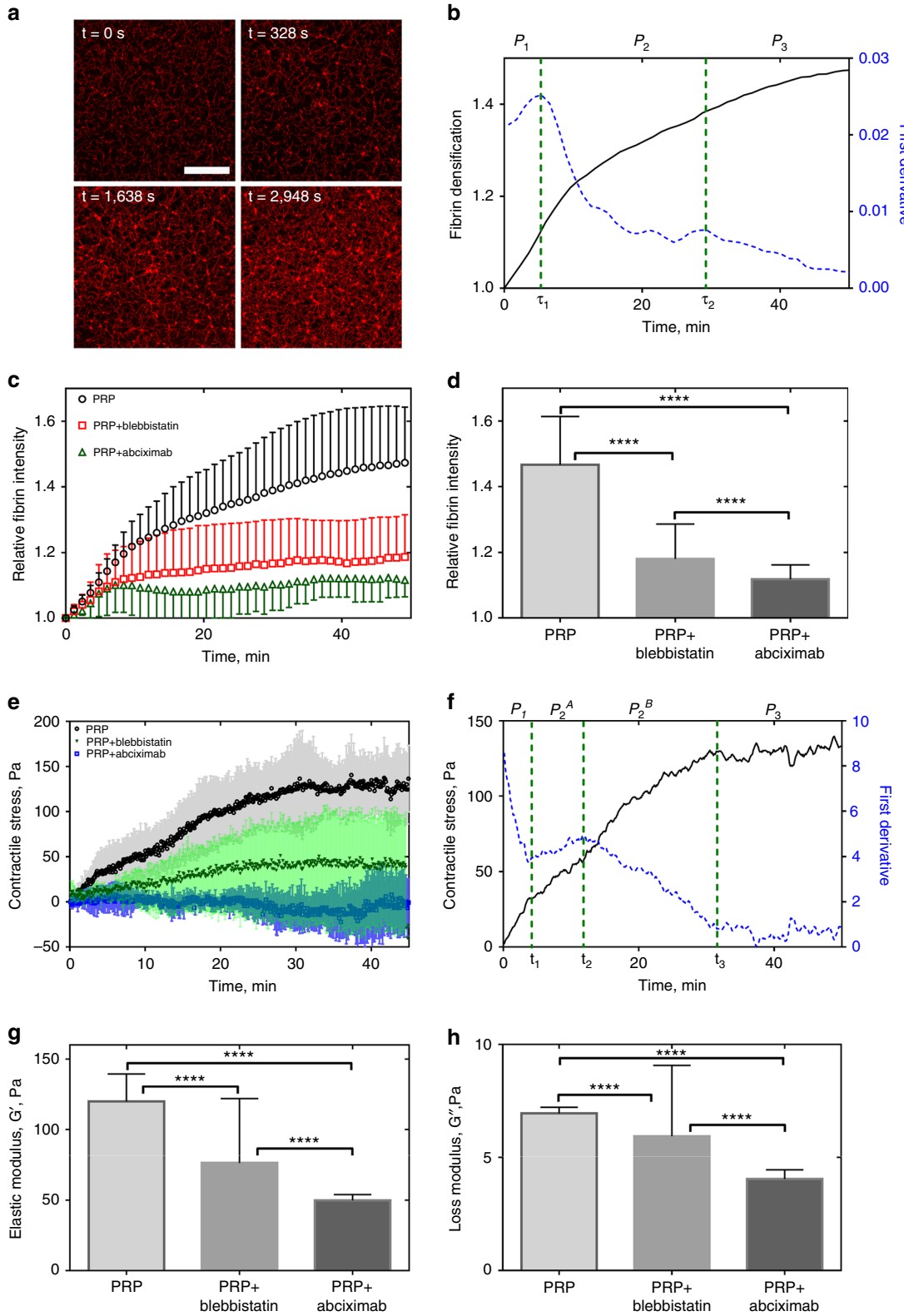

To examine the spatio-temporal non-uniformity in the clot contraction rate, clots were formed on a detergent-treated glass surface (Methods section) to prevent fibrin attachment and to allow for unconstrained movement/shrinkage of the clot during contraction. Time-lapse z-stack confocal imaging of the platelet-fibrin meshwork revealed drastic differences in the speed of translocating platelets at the edge of the contracting clot moving inwards and inside the clot (Fig. 6a, b). The mean speed of platelets near the edge of the clot was $0.020 \pm 0.018$ μm/s, while platelets in the clot interiors moved with an average speed of $0.009 \pm 0.006$ μm/s (Fig. 6c). The speed at the clot edge varied from 0.001 to 0.23 μm/s, while in the interior of the clot the speed ranged from 0.001 to 0.05 μm/s (Fig. 6d). Hence, spatial anisotropy of contracting clots is due to the faster moving edges and less mobile clot's interior domains. Such a difference in the speed of platelet displacement can be explained using a

phenomenological model of a contracting clot (Supplementary Note 2). According to the model, platelets remain stationary in the center of a contracting clot, while the rate of movement increases toward the edge of the clot. The quantitative characteristics of clot contraction are provided in Supplementary Table 1.

## Discussion

Contraction is a final stage of blood clot maturation that strengthens hemostatic clots, enhances wound healing and restores blood flow past otherwise obstructive thrombi. The physical mechanisms of clot contraction have remained unknown, although certain correlations between blood clot cellular composition and clot contraction mechanics were identified[1]. The existing tentative ideas of platelet-fibrin interactions

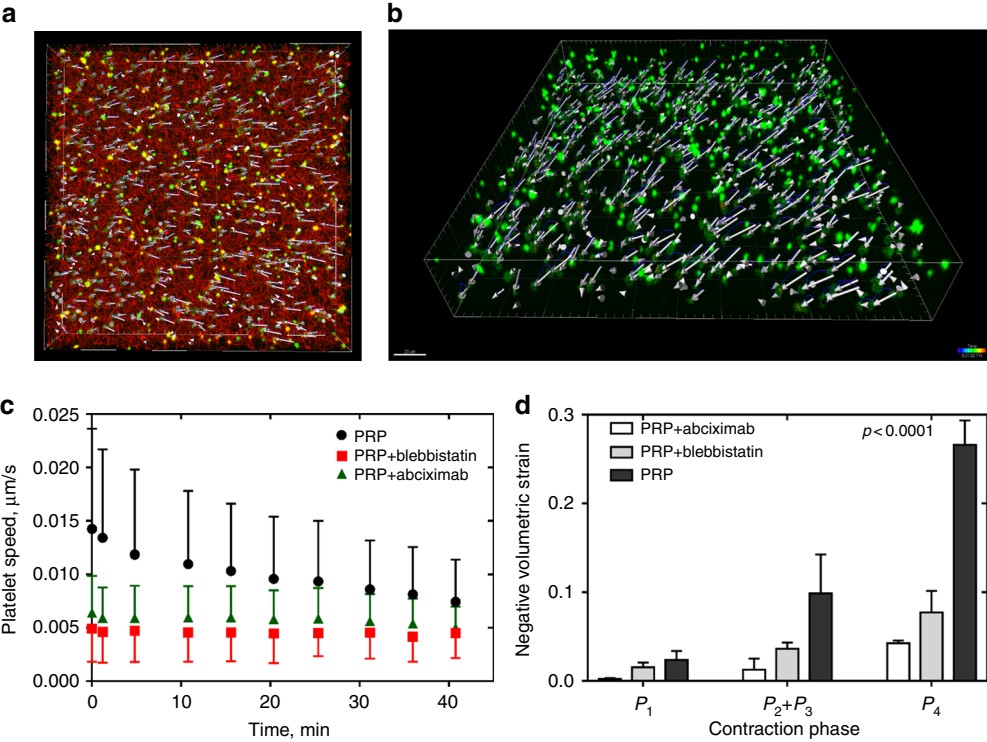

**Fig. 5** Deformation of the platelet-fibrin meshwork during contraction quantified using platelet displacement as a marker indicative of local strain.
**a**, **b** Representative images of a three-dimensional platelet displacement field of a contracting clot; **a** *top view*, fibrin (*red*), platelet (*green*), and platelet displacement vectors are shown; **b** *perspective view*, platelets (*green*) with their displacement vectors are visualized. The clot volume analyzed was 217 μm × 217 μm × 34 μm. **c**, **d** Changes of the platelet velocity during clot contraction **c** and clot volume reduction **d** in the absence and presence of blebbistatin (300 μM) and abciximab (100 μg/ml) (mean ± SD, 310 platelets analyzed in three clots). Statistical significance shown between the volumetric strains of contracted clots in the absence and presence of blebbistatin or abciximab; $P < 0.05$, Mann−Whitney U-test

**Fig. 4** Structural and mechanical contraction kinetics of PRP-clots. **a** Serial confocal images showing time-dependent densification of the fibrin network; *Scale bar*: 60 μm. **b** Dynamic fibrin fluorescence intensity per square micrometer as a measure of fibrin densification (*black solid line*) characterized by three phases determined by the local extremes of the first derivative (*blue dashed line* and *vertical dashed* borders between the phases). The phase designations shown here correspond to the phases $P_1–P_4$ shown in Fig. 1b. **c** Fibrin densification as a function of time in the absence and presence of blebbistatin (300 μM) and abciximab (100 μg/ml) (mean ± SD, $n = 3$). **d** Fibrin fluorescence intensity (fibrin density) at the end of contraction in the absence and presence of blebbistatin (300 μM) and abciximab (100 μg/ml) (mean ± SD, $n = 3$, ****$P < 0.0001$, a two-tailed Mann−Whitney test). **e** Dynamic contractile stress generated by the platelet-fibrin meshwork in the absence and presence of 100 μg/ml abciximab, a specific antagonist of the platelet receptor integrin αIIbβ3 (averaged kinetic curves, $n = 3$, mean ± SD). **f** The kinetic curve shown in **e** has four phases ($P_1$, $P_2^A$, $P_2^B$, and $P_3$) defined as in **b**. The first phase, $P_1$, corresponds to the pre-contraction stage with the rate constant $k_1$; $P_2^A$ and $P_2^B$ comprise the fastest phases with the rate constants $k_2$ and $k_3$, respectively; and $P_3$ is the final contraction phase with the rate constant $k_4$. **g**, **h** The storage and loss moduli of fully contracted PRP-clots (50 min) in the absence and presence of blebbistatin (300 μM) and abciximab (100 μg/ml) ($n = 3$, mean ± SD, ****$P < 0.0001$, a two-tailed Mann−Whitney test)

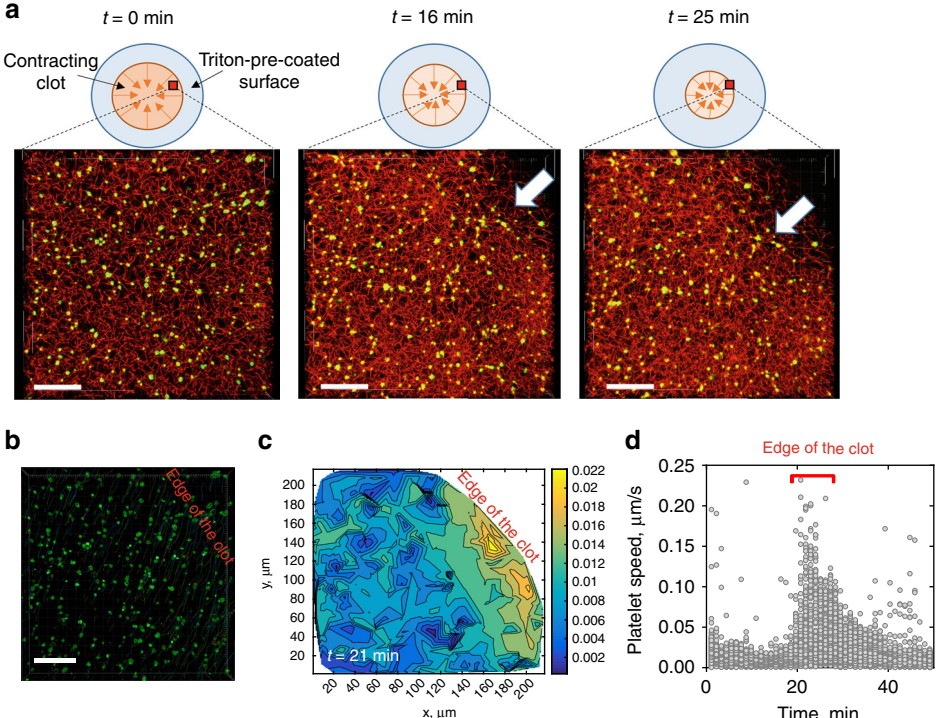

**Fig. 6** Non-uniform deformation of platelet-fibrin meshwork during clot contraction. Clots were formed on a Triton X-100 pre-coated surface to allow for unconstrained contraction of the clot edges in both vertical and horizontal directions. **a** Platelet-fibrin mesh *z*-projections are shown for three different time points during clot contraction and reveal inward movement of the clot edge (*white arrow*); scale bar: 40 μm. **b** Three-dimensional (3D) tracking of individual platelets embedded into the fibrin network allow visualization of the 3D platelet displacement field; *Scale bar*: 40 μm. **c** A representative contour plot of the mean platelet speed inside a clot (20 μm from the *bottom*) based on the reconstituted 3D platelet displacement field. The color bar shows the mean speed of platelet displacement (μm/s). **d** Changes of the platelet moving speed with time during clot contraction, indicating higher speed values at the edge of contracting clots (310 cells in three clots). Data presented in **d** show the absolute velocity of platelets identified in the temporal sequence of 3D images shown in **a**. For each time point in **d**, a set of data points shown corresponds to velocities of platelets tracked in a 3D volume of contracted clot provided in **a**. As the edge of the clot passes through the view area between 20 and 30 min time points, platelets reach their high velocity values corresponding to a yellow region in the velocity contour plot in **c**. For the entire sequence of events see Supplementary Movies 7–11

can be segregated into several conceptual groups. The first one is based on the assumption that platelets reach out with their filopodia to combine with and attach to fibrin fibers, pulling the fibrin mesh toward the platelets[24]. In the second scenario, filopodia are assumed to perform sweeping movements that gather in and compress fibrin fibers[25]. The third idea is based on platelet–platelet interactions via extended filopodia that join adjacent cells[26], after which the surface of the aggregating cells is zippered together, thus compressing fibrin fibers inside[26]. Until now there has been no convincing evidence for any of these scenarios, since they are based on static electron microscopy or light microscopy images, so the physical actions of the dynamic platelet-fibrin interplay could not be observed.

Our results reveal that extension-retraction of platelet filopodia is a basic function needed for transmitting platelet contractile force and re-arranging the fibrin matrix, which represents a novel function for filopodia, since they have generally been thought to be the cell's sensory and guiding organelles. These data are in agreement with previous studies suggesting that platelet filopodia are functionally important in (patho)physiological processes such as tissue healing, development, dorsal closure[27–31], phagocytosis[32], and cancer[33]. Yet, in contrast to our paper, none of these studies provided a cellular mechanism for these effects. In a more general context, the important physiological functions of cellular filopodia were shown to be necessary for tissue patterning, establishment of cell–cell junctions, haptotaxis, and matrix sensing[34].

Our study provides quantitative structural details of how contracting platelets cause clot volume shrinkage accompanied by dramatic structural alterations of the platelet-fibrin meshwork. We found that activated platelets bend and shorten individual fibrin fibers via their filopodia that undergo sequential extension and retraction, as if pulling hand-over-hand on a rope. Platelets also induce compaction of fibrin fibers into platelet-attached agglomerates. As a result of simultaneous pulling on multiple, closely spaced fibrin fibers, platelets pull themselves closer to each other and form secondary clusters larger than the initial aggregates. Contracting platelets actively remodel the fibrin network by increasing its density followed by enhanced clot stiffness. Kinetic analysis of the time course of structural and mechanical transitions revealed a multiphasic behavior with at least three distinct phases that differ in duration and rate constants. All the observed changes were reduced or abrogated in the presence of specific inhibitors of non-muscle myosin IIA (blebbistatin) and the platelet integrin $\alpha_{IIb}\beta$ (abciximab), indicating that actomyosin-driven cell contractility and integrin-fibrin mediated platelet-fibrin interactions are crucial for contraction of blood clots. Finally, blood clot contraction was found to be a spatially non-uniform process with faster compression of the clot edge and a delayed deformation of the clot interior. Altogether, the results provide a quantitative structural basis for the mechanobiology of clot contraction at various spatial scales from a single cell/single fiber level up to the network and macroscopic levels. Our results obtained on platelet-induced contraction of the filamentous fibrin network are of fundamental importance because they provide a

foundation for understanding the dynamic and complex biomechanical interplay between non-muscle cells and the fibrous extracellular matrices of various compositions.

## Methods

**Platelet-rich plasma.** Blood was drawn by venipuncture from healthy volunteers not talking aspirin or other medications known to affect platelet function for at least 10 days. Informed consent was obtained in accordance with a protocol approved by the University of Pennsylvania Institutional Review Board. Platelet-rich plasma (PRP) was prepared from whole blood drawn into 3.8% trisodium citrate (9:1 v/v). To obtain PRP, the citrated blood was centrifuged at $210 \times g$ at 25 °C for 15 min. The platelet count was performed using an automated hematology analyzer (HemaVet 950FS, Drew Scientific). PRP samples were kept at room temperature and studied within 4 h after blood collection.

**Contracting clots.** To visualize distinctly fibrin and platelets in a fluorescent confocal microscope, Alexa-Fluor 594-labeled human fibrinogen (Molecular Probes, Grand Island, NY, 40 μg/ml final concentration) and calcein green, AM (Molecular Probes, 0.2 μg/ml final) were added to PRP samples and incubated for 10 min at 37 °C before initiation of clotting. Clotting was induced by adding $CaCl_2$ (40 mM final concentration) and thrombin (0.75 or 1 U/ml final concentration) to PRP containing the fluorescent probes. Thirty microliters of samples was quickly transferred onto $35 \times 10$ mm PELCO clear wall glass bottom cell culture dishes in the environmental chamber of the confocal microscope for the time course z-stack imaging. In some experiments, the glass was pre-coated with 4% (v/v) Triton X-100 in PBS to prevent attachment of fibrin and allow for unconstrained clot contraction. In inhibitory experiments abciximab (ReoPro; Eli Lilly, Indianapolis, IN, USA) was added to PRP at 100 μg/ml or blebbistatin (Sigma-Aldrich, St. Louis, MO, USA) was used at 300 μM (both final concentrations) prior to the fluorescent probes and incubated for 15 min at 37 °C.

**Confocal microscopy of contracting clots.** Contracting clots were imaged in a Zeiss LSM710 laser confocal microscope with Plan Apo ×40 (NA 1.2) water immersion objective lens to obtain a series of high-resolution 35 μm-thick z-stack images taken during 30–50 min over the course of clot formation and contraction. The distance between each z-stack slice was 0.8 μm with a resolution of $1024 \times 1024$ pixels. At the end of each clot contraction experiment, a 212.5 μm × 212.5 μm × 103 μm z-stack was taken with a voxel size of $0.2076 \times 0.2076 \times 0.6896$ μm$^3$. A combination of an argon laser beam with a 488-nm wavelength and a helium-neon laser beam with a 594-nm wavelength was utilized for confocal microscopy of fluorescently labeled platelets and fibrin, respectively.

**Image analysis and data presentation.** The images of contracting clots obtained with a confocal microscope were analyzed with a number of complementary software packages to characterize the clots' structural dynamics. Changes in fibrin fluorescent intensity, its density and co-localization of fibrin and platelets were analyzed using the open source Fiji software[35]. Cubic convolution interpolation using the TransformJ plugin in the Fiji software was applied to the original confocal data to acquire zoomed-in images of platelet filopodia and individual fibrin fibers shown in Fig. 1c, d. The length measurements presented in Fig. 1b were based on the original confocal microscopy images using a standard distance measurements segmented-line tool in Fiji.

The amount of fibrin associated with individual platelets was quantified by measuring the fibrin area co-localized with a platelet, using a standard co-localization analysis of the Fiji software. Following acquisition of a co-localized pixel map and platelet-co-localized fibrin segmented images (Fig. 2a, right inset), the area of compacted fibrin fragments was measured using a standard Fiji analysis tool named "Analyze Particles".

The fraction of fibrin compacted by platelet aggregates was calculated as the ratio of the total area of compacted fibrin co-localized with platelets having area > $S^{SP}$ to the overall fibrin amount associated with both single platelets and platelet aggregates.

Platelets were tracked using Imaris 8.2, visualization and 3D reconstruction were done in Fiji, Imaris and Volocity, and numerical and graphical data analysis and presentation were done in MATLAB and GraphPad Prism 6.

**Rheometry of contracting clots.** We used a rotational rheometer (ARG2; TA Instruments, New Castle, DE, USA) to quantify changes in viscoelastic properties of PRP-clots during contraction over the time course of 50 min. A 20-mm flat parallel top rotating plate and a fixed bottom plate were employed as the test geometry. PRP samples were activated with $CaCl_2$ and thrombin as described above, placed in a 400 μm-gap between the top and bottom plates and a low oscillatory shear strain (1%, 1 Hz) was applied. The elastic and viscous properties of the clot were characterized by their storage ($G'$) and loss ($G''$) moduli, respectively. In addition, the contractile force generated by the platelet-fibrin matrix was measured as a negative normal (pulling) force applied to the rheometer plates. To prevent sample drying at the clot-air interface, a layer of silicone oil was applied at the edge of the clot.

**Analysis of clot contraction kinetics.** Kinetic analysis was done by calculating the first derivative of the curves reflecting changes of fibrin fluorescence intensity over time and normal force curves. Peaks of the first derivative enabled us to segregate the kinetic curves into distinct contraction phases characterized by discrete rate constants. Curve fitting was done using a polynomial function over the time course of clot contraction.

**Statistical analysis.** For clot contraction assays, statistical analysis using 40 z-stacks of confocal microscopic images for a single clot sample in parallel multiple sample measurements was conducted. All data were obtained from at least three independent measurements and were expressed as mean ± SD or mean ± SEM as indicated. Additional information about the platelet sample size is provided in the Supplementary Table 1. Statistical significance (defined as $P < 0.05$) was calculated using a two-sided Mann–Whitney U-test.

**Data availability.** All relevant data are available from the authors.

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

## Acknowledgements

We would like to acknowledge support from the National Institutes of Health grants NIH U01HL116330 (O.V.K., R.I.L., J.W.W., and M.S.A.) and HL090774 (R.I.L. and J.W.W.), Scientist Development Grant from American Heart Association 17SDG33680177 (O.V.K.), Walter Cancer Foundation (O.V.K.), the Program for Competitive Growth at Kazan Federal University (R.I.L.), NSF grant DMR 1505662 (J.W.W. and R.I.L.) and grant EPSRC EP/C513037/1 to P.R. Williams (Swansea University, Wales, UK) for the TA Instruments ARG2 rheometer.

## Author contributions

Conceptualization, O.V.K., R.I.L., and J.W.W.; Formal Analysis, O.V.K. and R.I.L.; Investigation, O.V.K. and R.I.L.; Resources, M.S.A. and J.W.W.; Writing—Original Draft: all authors; Supervision, M.S.A. and J.W.W.; Project Administration, M.S.A. and J.W.W.; Funding Acquisition: M.S.A. and J.W.W.
