## [Peer Review File · Nature Communications]

Reviewers' Comments:

Reviewer #1:

This is an interesting article and it is worthy of publication. There are a few small points that should be addressed. 1. Phases in the contraction are poorly defined and breaks in a curve of rate of contraction or force generation do not necessarily constitute a functional change. Thus, it is presumptuous to describe these as phase differences unless there is a clear change in the functional behavior of the system (different mode of contraction or actomyosin organization). If the number of filopodia extended or the mode of movement changed, that would strengthen the idea of distinct phases. 2. Figure 1 does show changes in platelet shape but filopodia are not clearly seen. It would be helpful to show a high mag view of a platelet filopodium and its contraction.

Reviewer #3:

Remarks to the Author:

The study has a number of interesting, novel, and important observations that have direct bearing on our thinking of how platelets regulate blood clot formation, stabilization, and contraction.

The authors make a compelling case for the platelets shortening fibrin strands via a repeated sequence of forming kinks and accumulating fibrin on their surfaces. It's a remarkable observation they've made showing platelets actively remodeling and contracting their local fibrin network. The author's 'hand over hand' conceptualization is well supported by the images and greatly clarified a fairly murky area of platelet behavior in 3-D clots, as opposed to their very well characterized behavior on flat, mostly glass surfaces.

What has long been a point of confusion regarding the role of platelets in clot formation and eventual contraction has been how the platelet's own cellular contraction might lead to the macroscopic reductions in clot size of 25 to 50% or more (easily visible in a glass tube). Platelets themselves can only contract a tiny amount (a few percent, allowing for excess membrane area). The authors in this study present compelling visual and quantitative information supporting their conceptual model that makes the platelet role in clot contraction much more intelligible.

That noted, the manuscript figures in several instances are hard to interpret. The majority of my comments are on readability and intelligibility of the figures.

Minor Comments.

The spatial resolution of the imaging approach isn't given consistently, through there are scale bars present. As the pixel size is 0.207 microns it seems that a comment on the spatial resolution is warranted. To clarify, are the lengths in the Figure 1B histogram interpolated? The binning that is evident suggests this may be so.

If the clot volume is 35 microns deep, at what distance from the glass surface are the bulk of the displayed images taken? It seems that it might be possible that there was more contraction at the midpoint between the coverslips than on either glass surface, where there might be some fibrin-platelet-glass interactions.

Figure 1A, 'platelet' misspelled in image insert.

Figure 2: The explanation of how the fibrin 'compaction' measurement is made is unclear. Furthermore, how is the amount of fibrin normalized? The image doesn't indicate where the other 'compacted' fibrin is located or how it is defined relative to the 'uncompacted' fraction. If ultimately 30% of the compacted fibrin is associated with the platelet aggregates or platelet surface, then it seems the other 70% of the compacted fibrin would not be platelet associated--is this a correct inference?

In Figure 2D, as a point of clarification, the amount of fibrin associated with the platelet surface is approximately 10 times the amount of fibrin not associated with the platelet or platelet aggregates. Does this mean that the 'uncompacted' fibrin is what is visible as fibrin strands?

In the E and D panels, it appears that the PRP (untreated) clot is compacted, hence more fibrin should be in the imaged area. Is the average fluorescence increased in the compacted clot?

The y-axis for Figure 3B is difficult to understand. The axis indicates a long scale of platelet concentration (based on the original blood draw?) versus the 2-D projection of the area of the platelet cluster. Isn't there a more direct way to convey the fact that the number of platelets in a clump doesn't seem to be changing much while there are major changes in the fibrin compaction? The units on the insert of Figure 3B do not seem consistent with the y-axis on the major plot.

In line 144 the comment is made that secondary enlargement of platelet clusters likely results in generation of a higher contractile force for later clot contraction. While there is clearly a correlation, it is also not clear to me that this follows directly. At some point in the text the authors almost appear to argue the opposite as they discuss why their estimates of platelet contractile force are much lower than measures made in AFM or by traction force microscopy of individual platelets. By their model, the aggregated platelets may well be pulling against each other and therefore not generating a greater traction force in any specific direction. So it isn't straightforward to conclude larger aggregates generate more traction force.

Figure 4 is an impressive demonstration of the role of the platelet actinomyosin machinery in the fibrin compaction process. But in Figure 4e why is there no data for blebbistatin, since that appears to be the key test?

What do the four phases of contraction in 4E represent in terms of the biology of the system? Are different platelet signaling mechanisms being activated or is the fibrin being organized in different ways at each phase? It seems possible that the observation of four different phases, in the absence of any mechanistic data, may be a pattern unique to the specific geometry or shape of the clot in the confocal microscope.

In Figure 6D, are these the same platelets that can be identified in the time sequence of Figure 6A? It is not clear how the time and spatial elements are integrated in this figure.

We would like to thank the reviewers for their careful evaluation of our manuscript. We are very pleased that the reviewers have found our studies to be interesting and worthy of publication, and novel and important. We are especially glad that they consider the observations remarkable and compelling visual and quantitative information. At the same time, the reviewers had several good suggestions and valid concerns, especially relating to some aspects that were not clear. We have addressed all the comments of the reviewers as specified in detail below, and revised our manuscript accordingly. These critiques and comments have helped us to improve and strengthen the manuscript considerably. The changes to the manuscript are highlighted in red. Itemized responses to each of the reviewer's comments are summarized below.

Reviewer #1

This is an interesting article and it is worthy of publication. There are a few small points that should be addressed.

Response:

We thank this reviewer for the positive evaluation.

Comment 1. Phases in the contraction are poorly defined and breaks in a curve of rate of contraction or force generation do not necessarily constitute a functional change. Thus, it is presumptuous to describe these as phase differences unless there is a clear change in the functional behavior of the system (different mode of contraction or actomyosin organization). If the number of filopodia extended or the mode of movement changed, that would strengthen the idea of distinct phases.

Response:

We previously demonstrated through accurate measurement of macroscopic changes in the volume of the contracting clot as a function of time that clot contraction is comprised of three sequential phases, each characterized by a distinct rate constant (Tutwiler, et al., Blood 2106). The three phases of this process represent: Initiation of Contraction, Linear Contraction, and Mechanical Stabilization. We showed how various processes involving platelets, fibrin, and/or RBCs impact the procession of clot contraction through the three phases. What is remarkable is that the same three phases are present at the microscopic level in the results of this paper. Our data showing time-dependent changes in the shortening rate of platelet filopodia (Fig 1B, inset, right y-axis) support the idea of alterations in the functional behavior of the system, underlying the phase differences. There are several peaks of the first derivative curve in the figure, suggesting the existence of different modes of the filopodia contraction dynamics. The time borders between these modes at 4 min and 14 min correspond to the temporal borders of the network structure-based and the whole-clot-mechanics-based phases of contraction. Therefore, the distinct phases of contraction likely reflect time-dependent functional changes in the course of contraction that are yet not fully understood.

We have added the following text in the section "Kinetics of clot contraction based on the dynamics of contractile stress" on page 9:

"Remarkably, the single-fiber contraction rates (Fig. 1B, inset) signify local minima around $t^=4$ min and $t^{**}=14$ min, which correspond to the P_1 - P_2 and P_2^A - P_2^B phase transition times obtained*

from macroscopic whole clot kinetics measurements (Tutwiler, et al., Blood 2016). Because platelet contraction and fibrin network remodeling are the key elements of the entire process of clot contraction, the observed correspondence between the phases determined at the micro- and macro-scales suggests that the multiphasic behavior observed at the whole clot level originates from the time-dependent functional mechanisms at the level of single platelet/single fiber interactions.”

Figure 1 legend is updated accordingly by adding the following sentence: “ t^ , t^{**} are the microscopic phase transition times separating different regimes of filopodia shortening.”*

Comment 2. Figure 1 does show changes in platelet shape but filopodia are not clearly seen. It would be helpful to show a high mag view of a platelet filopodium and its contraction.

Response:

In response to the reviewer’s comment, we have provided a time series of high-magnification and higher resolution images of an individual platelet filopodium undergoing contraction (Supplementary Video S2A in the supplementary materials).

*Figure 1 legend is update accordingly by adding the following clarification in red: “(see **Supplementary Video S2** for the full sequence and the high magnification **Supplementary Video S2A** showing an individual platelet filopodium undergoing contraction)”*

Reviewer #3

The study has a number of interesting, novel, and important observations that have direct bearing on our thinking of how platelets regulate blood clot formation, stabilization, and contraction.

The authors make a compelling case for the platelets shortening fibrin strands via a repeated sequence of forming kinks and accumulating fibrin on their surfaces. It’s a remarkable observation they’ve made showing platelets actively remodeling and contracting their local fibrin network. The author’s ‘hand over hand’ conceptualization is well supported by the images and greatly clarified a fairly murky area of platelet behavior in 3-D clots, as opposed to their very well characterized behavior on flat, mostly glass surfaces.

What has long been a point of confusion regarding the role of platelets in clot formation and eventual contraction has been how the platelet’s own cellular contraction might lead to the macroscopic reductions in clot size of 25 to 50% or more (easily visible in a glass tube). Platelets themselves can only contract a tiny amount (a few percent, allowing for excess membrane area). The authors in this study present compelling visual and quantitative information supporting their conceptual model that makes the platelet role in clot contraction much more intelligible.

Response:

We thank this reviewer for such a positive evaluation of our work, both in terms of remarkable observations and new concepts important to understand clot contraction.

General Comment

That noted, the manuscript figures in several instances are hard to interpret. The majority of my

comments are on readability and intelligibility of the figures.

Response:

We have responded below in detail to all instances concerning the readability and intelligibility of the figures. We greatly appreciate all of these suggestions because we want the illustrations of our observations to be as clear as possible.

Minor Comments.

Comment 1. The spatial resolution of the imaging approach isn't given consistently, though there are scale bars present. As the pixel size is 0.207 microns it seems that a comment on the spatial resolution is warranted. To clarify, are the lengths in the Figure 1B histogram interpolated? The binning that is evident suggests this may be so.

Response:

In response to the reviewer's comment, we have added information on image resolution when the zoomed-in images of individual platelets and fibers were presented. In particular, the following text was added to Materials and Methods, a subsection entitled "Image analysis and data presentation" (page 15): "Cubic convolution interpolation using the TransformJ plugin in the Fiji software was applied to the original confocal data to acquire zoomed-in images of platelet filopodia and individual fibrin fibers shown in Fig 1C,D."

At the same time, the length measurements presented in Fig 1B were done based on the original confocal microscopy images without interpolation using a standard "Distance measurements segmented-line tool" in Fiji. With this regard, the following sentence was added into Materials and Methods on page 15: "The length measurements presented in Fig 1B were based on the original confocal microscopy images using a standard distance measurements segmented-line tool in Fiji".

Comment 2. If the clot volume is 35 microns deep, at what distance from the glass surface are the bulk of the displayed images taken? It seems that it might be possible that there was more contraction at the midpoint between the coverslips than on either glass surface, where there might be some fibrin-platelet-glass interactions.

Response:

The bulk of the displayed images during clot contraction dynamics were taken at 10 microns from the glass surface. It is noteworthy that clot contraction was observed under non-isometric (unconstrained) conditions when the clot was formed on a pre-lubricated glass surface to prevent attachment of clots, to ensure that there are no surface effects. In response to the reviewer's comment, we have provided additional 100-um thick high-resolution 3D z-stack images of a contracted clot (Figure FS1 in the supplementary materials) showing no major differences in the density of a fibrin network throughout the clot volume. The Figure FS1 caption was updated accordingly with the additions shown in red.

Figure FS1. A, B: Representative confocal images of a platelet-fibrin meshwork in a PRP-clot formed and allowed to contract in the presence of abciximab (100 $\mu\text{g/ml}$). Scale bar is 30 μm . **C:** A xy-projection of a fibrin z-stack confocal images in a fully contracted PRP-clot, **D:** changes in fibrin fluorescence intensity with distance from the bottom of the clot, corresponding to C and showing no significant differences in the density of a fibrin network throughout the clot volume.

Comment 3. Figure 1A, ‘platelet’ misspelled in image insert.

Response:

We have corrected the misspelled label in Figure 1A to “platelet+fibrin”

Comment 4. Figure 2: The explanation of how the fibrin ‘compaction’ measurement is made is unclear. Furthermore, how is the amount of fibrin normalized? The image doesn’t indicate where the other ‘compacted’ fibrin is located or how it is defined relative to the ‘uncompacted’ fraction. If ultimately 30% of the compacted fibrin is associated with the platelet aggregates or platelet surface, then it seems the other 70% of the compacted fibrin would not platelet associated--is this a correct inferences?

This comment has been segregated into sub-comments 4a-4d.

Sub-Comment 4a. Figure 2. The explanation of how the fibrin ‘compaction’ measurement is made is unclear.

Response:

In response to the reviewer’s comment, we have provided additional details on how the fibrin ‘compaction’ measurements were made by adding the following text in the Materials and Methods section on pages 15,16:

“The amount of fibrin associated with individual platelets was quantified by measuring the fibrin area colocalized with a platelet, using a standard colocalization analysis of the Fiji software. Following acquisition of a colocalized pixel map and platelet-colocalized fibrin segmented images (Figure 2A, right inset), the area of compacted fibrin fragments was measured using a standard Fiji analysis tool named “Analyze Particles”.”

Sub-Comment 4b. Furthermore, how is the amount of fibrin normalized?

Response:

The amount of fibrin was normalized by its initial amount at $t=0$ (pre-contraction phase) by dividing the fibrin fluorescence intensity at the time point t by the fibrin fluorescence at $t=0$.

Sub-Comment 4c. The image doesn't indicate where the other 'compacted' fibrin is located or how it is defined relative to the 'uncompacted' fraction.

Response:

We have provided additional clarification of the terms 'compacted' vs. 'uncompacted' fibrin. We refer to the fibrin patches associated with platelets pulling on or actively contracting fibrin fibers to as 'compacted' fibrin, while the rest of platelet-colocalized fibers prior to contraction are defined as 'uncompacted' fibrin.

To make the interpretation of Figure 2D more clear, we have corrected the y-axis as "Normalized Area of Platelet-Colocalized Fibrin" in Figure 2D. Normalization was done by dividing the fluorescence intensity of fibrin in a contracted clot by the fluorescence of fibrin in a pre-contracted clot ($t=0$).

We have updated Figure 2 legend accordingly as

"(B, C): Changes in the total area of platelet-associated fibrin normalized to its initial value (B) and the mean absolute area of platelet-colocalized fibrin (C)".

Sub-Comment 4d. If ultimately 30% of the compacted fibrin is associated with the platelet aggregates or platelet surface, then it seems the other 70% of the compacted fibrin would not platelet associated--is this a correct inferences?

Response:

We have clarified the title of the y-axis to make the interpretation of the data clearer. We distinguish fibrin compaction by individual platelets and platelet aggregates. The fraction of fibrin compacted by platelet aggregates was calculated as a ratio of the total area of compacted fibrin colocalized with platelets (having an area $>S^{sp}$) and the overall amount of fibrin associated with both single platelets and platelet aggregates. Thus, Figure 2D shows changes in the ultimate fraction of fibers compacted by platelet aggregates which achieves 30% in the fully contracted clot, while the other 70% of the compacted fibers is associated with single platelets.

In response to the reviewer's comment, we have reorganized the panels in Figure 2 to correspond the order mentioned in the text.

Comment 5. In Figure 2D, as a point of clarification, the amount of fibrin associated with the platelet surface is approximately 10 times the amount of fibrin not associated with the platelet or platelet aggregates. Does this mean that the 'uncompacted' fibrin is what is visible as fibrin strands?

Response:

This is correct and the uncompacted fibrin is visible as free fibrin strands or fibers. For the sake of clarification, we would like to emphasize that Fig 2D shows the amount of fibrin accumulated by platelets and platelet aggregates (compacted fibrin) relative to the initial amount of fibrin associated with platelets prior to contraction.

Comment 6. In the E and D panels, it appears that the PRP (untreated) clot is compacted, hence more fibrin should be in the imaged area. Is the average fluorescence increased in the compacted clot?

Response:

It seems that the reviewer meant panels E and F. This is correct. There was more fibrin in the imaged area that appeared in the course of PRP clot contraction than in the case of blebbistatin-treated clots in which contraction was impeded. Clot contraction was accompanied by an increase in the average fluorescence intensity, as quantified in detail and shown in Figure 3A-C.

Comment 7. The y-axis for Figure 3B is difficult to understand. The axis indicates a long scale of platelet concentration (based on the original blood draw?) versus the 2-D projection of the area of the platelet cluster. Isn't there a more direct way to convey the fact that the number of platelets in a clump doesn't seem to be changing much while there are major changes in the fibrin compaction? The units on the insert of Figure 3B do not seem consistent with the y-axis on the major plot.

Response:

Indeed, the units on the insert of Figure 3B do not seem consistent with the y-axis in the major plot. In response to this comment, we have corrected the units in the inset for Figure 3B accordingly.

Comment 8. In line 144 the comment is made that secondary enlargement of platelet clusters likely results in generation of a higher contractile force for later clot contraction. While there is clearly a correlation, it is also not clear to me that this follows directly. At some point in the text the authors almost appear to argue the opposite as they discuss why their estimates of platelet contractile force are much lower than measures made in AFM or by traction force microscopy of individual platelets. By their model, the aggregated platelets may well be pulling against each other and therefore not generating a greater traction force in any specific direction. So it isn't straightforward to conclude larger aggregates generate more traction force.

This comment has been segregated into sub-comments 8a and 8b.

Sub-Comment 8a. In line 144 the comment is made that secondary enlargement of platelet clusters likely results in generation of a higher contractile force for later clot contraction. While there is clearly a correlation, it is also not clear to me that this follows directly.

Response:

We agree with the reviewer that the secondary enlargement of platelet clusters leading to generation of higher force at the later steps of clot contraction is not evident, so we have changed this sentence. Instead, the major point here is that our data indicate that larger mass of fibrin is accumulated on platelet aggregates, suggesting that platelet aggregation enhances remodeling of fibrin matrix leading to formation of larger clumps of densified (or compacted) fibrin as compared to amount of fibrin compacted by single platelets. Formation of larger pieces of compacted fibrin means that platelet aggregates performed more work on pulling and compacting fibrin fibers than individual platelets, although not necessarily generating higher contractile forces.

Sub-Comment 8b. At some point in the text the authors almost appear to argue the opposite as they discuss why their estimates of platelet contractile force are much lower than measures made in AFM or by traction force microscopy of individual platelets.

Response:

There are at least two reasons which might explain why our estimates of the force developed by a platelet in a fibrin matrix is lower than the platelet pulling force measured by AFM or in traction force microscopy experiments. First, there is a difference in the dimensionality: during the AFM or traction force microscopy experiments the platelet pulling force is probed along one direction, whereas in a 3D fibrin network platelets pull on multiple fibers therefore distributing the contractile force in all directions. Furthermore, as the reviewer suggests, the platelets may be working against each other. Second, mechanical characteristics of fibrin fibers, such as bending rigidity and tensile stiffness, is another factor that might impact the resultant force as platelets in a clot, unlike an individual isolated platelet, need to perform work and dissipate energy on fiber deformations, including stretching, bending, and shortening of fibers, thus resulting in a lower estimate of the contractile force per platelet.

To make this point clearer, we have added the following text in Results section in page 9:

“As the maximum size of a compacted fibrin bundle is determined by the size of an associated platelet aggregate, our findings suggest that platelet aggregation enhances remodeling of fibrin matrix and results in formation of larger clumps of densified (or compacted) fibrin as compared to fibrin accumulated by single platelets. Since formation of compacted fibrin is driven by platelet actomyosin contractile machinery, a platelet aggregate dissipates more energy on pulling and compacting fibrin than an individual platelet by deforming larger amount of fibrin fibers.”

Comment 9. Figure 4 is an impressive demonstration of the role of the platelet actomyosin machinery in the fibrin compaction process. But in Figure 4e why is there no data for blebbistatin, since that appears to be the key test?

Response:

In response to the reviewer’s comment, we have obtained and added new data to Figure 4E showing a substantial reduction of the normal stress developed during clot contraction in the presence of blebbistatin.

Comment 10. What do the four phases of contraction in 4E represent in terms of the biology of the system? Are different platelet signaling mechanisms being activated or is the fibrin being organized in different ways at each phase? It seems possible that the observation of four different phases, in the absence of any mechanistic data, may be a pattern unique to the specific geometry or shape of the clot in the confocal microscope.

Response:

We previously demonstrated through accurate measurement of macroscopic changes in the volume of the contracting clot as a function of time that clot contraction is comprised of three sequential phases, each characterized by a distinct rate constant (Tutwiler, et al., Blood 2106). The three phases of this process represent: Initiation of Contraction, Linear Contraction, and Mechanical Stabilization. We showed how various processes involving platelets, fibrin, and/or RBCs impact the procession of clot contraction through the three phases. Thus, these three

phases are not unique to the geometry or shape. What is remarkable is that the same three phases are present at the microscopic level in the results of this paper. In addition, we were able to distinguish the fourth phase at the macroscopic level likely corresponding to exhaustion and profound dysfunction of platelets. We agree with the reviewer that the observed bulk kinetics response in contracted clots should be related to changes in biological functioning of the system. Although the details of the processes may be affected by clot geometrical or shape factors, the observed kinetics are mostly governed by platelet cytoskeletal reorganization and interaction of platelet filopodia with fibrin fibers rather than being a pattern corresponding to the specific shape of the clot at the macroscale. This is supported by the fact that multiphasic clot contraction kinetics was also observed for the entirely different clot contraction geometry (Tutwiler, et al., Blood 2106), being characterized in terms of phases similar to those identified in our study. A similar question was raised by Reviewer #1 and we have fully addressed this issue in the response to Comment 2 by Reviewer #1.

Comment 11. In Figure 6D, are these the same platelets that can be identified in the time sequence of Figure 6A? It is not clear how the time and spatial elements are integrated in this figure.

Response:

In response to the reviewer comment, we have added the following explanation in the legend of Figure 6 of the revised manuscript.

Data presented in D show the absolute velocity of platelets identified in the temporal sequence of 3D images shown in A. For each time point in D, a set of data points shown corresponds to velocities of platelets tracked in a 3D volume of contracted clot provided in A. As the edge of the clot passes through the view area between 20 min and 30 min time points, platelets reach their high velocity values corresponding to a yellow region in the contour velocity plot in C.

Reviewers' Comments:

Reviewer #1:

Remarks to the Author:

The manuscript has been significantly improved and I recommend acceptance.

We would like to thank the reviewers for their careful evaluation of our manuscript and recommendation of acceptance.
No actions were made as all the reviewers' comments were fully addressed.

REVIEWERS' COMMENTS:

Reviewer #1 (Remarks to the Author):

The manuscript has been significantly improved and I recommend acceptance.